# Adversarial Representation Learning for Hyperspectral Image Classification with Small-Sized Labeled Set

Shuhan Zhang [1], Xiaohua Zhang [1,*], Tianrui Li [1], Hongyun Meng [2], Xianghai Cao [1] and Li Wang [1]

1 School of Artificial Intelligence, Xidian University, Xi'an 710071, China; shzhang_66@stu.xidian.edu.cn (S.Z.); trli@stu.xidian.edu.cn (T.L.); caoxh@xidian.edu.cn (X.C.); liwang_233@stu.xidian.edu.cn (L.W.)
2 School of Mathematics and Statistics, Xidian University, Xi'an 710071, China; menghy@xidian.edu.cn
* Correspondence: xh_zhang@mail.xidian.edu.cn

**Abstract:** Hyperspectral image (HSI) classification is one of the main research contents of hyperspectral technology. Existing HSI classification algorithms that are based on deep learning use a large number of labeled samples to train models to ensure excellent classification effects, but when the labeled samples are insufficient, the deep learning model is prone to overfitting. In practice, there are a large number of unlabeled samples that have not been effectively utilized, so it is meaningful to study a semi-supervised method. In this paper, an adversarial representation learning that is based on a generative adversarial networks (ARL-GAN) method is proposed to solve the small samples problem in hyperspectral image classification by applying GAN to the representation learning domain in a semi-supervised manner. The proposed method has the following distinctive advantages. First, we build a hyperspectral image block generator whose input is the feature vector that is extracted from the encoder and use the encoder as a feature extractor to extract more discriminant information. Second, the distance of the class probability output by the discriminator is used to measure the error between the generated image block and the real image instead of the root mean square error (MSE), so that the encoder can extract more useful information for classification. Third, GAN and conditional entropy are used to improve the utilization of unlabeled data and solve the small sample problem in hyperspectral image classification. Experiments on three public datasets show that the method achieved better classification accuracy with a small number of labeled samples compared to other state-of-the-art methods.

**Keywords:** adversarial representation learning (ARL); generative adversarial networks; hyperspectral image

## 1. Introduction

Hyperspectral remote sensing technology uses a large number of spectral bands to image surface objects, these spectral bands are composed of ten to hundreds non-overlapping narrow bands and it combines spatial information with its own unique spectral information to acquire hyperspectral images on a pixel-by-pixel basis. An important research branch of hyperspectral remote sensing technology is HSI classification technology and it is the task of assigning categories to each pixel in a hyperspectral image. At present, this technology is widely used in many fields [1] such as modern agriculture [2], aviation [3], mineral exploration [4], astronomy [5], biomedicine [6], national defense, and homeland security monitoring [7].

For hyperspectral image classification techniques, the early studies only used the spectral information in hyperspectral datasets but ignored the spatial information that was contained in hyperspectral datasets, such as k-nearest neighbor (KNN) classifier [8], support vector machine (SVM) [9], distance classifiers [10], and extreme learning machines [11], etc. As the spectrum of pixel is affected by the surrounding pixel and noise, the algorithm using spectral information alone cannot obtain good classification performance. In subsequent studies, many researchers have introduced spatial information and formed the

classification method of HSI that is based on spatial-spectral combination. For example, the spatial-spectral composite kernel function method that is based on SVM [12]. This method combines hidden Markov random field segmentation with SVM [13]. The non-local weighted joint sparse representation classification method [14], and the joint sparse representation classification method that is based on shape adaptation [15]. Although the above methods have achieved some improvement in classification performance, their features are designed manually.

Deep learning is able to extract features from data automatically compared to ordinary machine learning that requires manual feature design, so many researchers have proposed HSI classification algorithms that are based on deep learning. For example, a stacked autoencoder [16] is used to extract hyperspectral spatial-spectral joint features [17]. Deep belief network (DBN) [18] is used to extract both spectral and spatial features of hyperspectral images and then cascade them to form spatial-spectral features for classification [19]. Convolutional neural network (CNN) [20] has features such as local perception, weight sharing, and multiple convolutional kernels, and thus has received attention from many researchers and many algorithms have been proposed [21]. For example, 1D-CNN [22] and 2D-CNN [23] are used to extract both spectral and spatial features, respectively. 1D-CNN and 2D-CNN are also combined to extract joint spatial-spectral features [24]. 3D-CNN is used to extract features directly from image blocks [25]. Pixel-pair convolutional neural network approach (PPF-CNN) [26] extends the training dataset by reorganizing and rescaling the training samples. However, the labeled samples of HSI are difficult to obtain and the training of deep learning requires a huge number of labeled samples data.

For solving the small sample problem, some scholars have proposed the use of transfer learning (TL) to classify hyperspectral images. For example, Lin et al. proposed an unsupervised classification method that was based on sample migration by transforming the source domain samples and target domain samples into the feature subspace [27], finding the transformation matrix between them and obtaining the initial classification results. Persello et al. used a feature migration strategy to reduce the difference between the source and target domain data by projecting the source domain data into the regenerative Hilbert space and calculate the distance between the conditional distributions of the source and target domains [28]. Some scholars have proposed some semi-supervised approaches, for example, Sun Z. proposed a semi-supervised support vector machine algorithm [29], which uses both labeled and unlabeled data to find a classification surface with maximum class spacing. Aydemir M. S. et al. proposed a graph-based HSI classification algorithm [30], which assumes that similar input data should share similar output labels, the vertices of the graph are labeled or unlabeled training samples, and the weights of the edges are the similarity between examples, followed by specifying the objective function to be optimized and using the smoothness as a regular term to find the optimal parameters. However, the distribution of unlabeled data that is used by the TL methods does not match the distribution of real samples and manual domain adaptation operations cannot completely solve this problem.

For the above problems, scholars have applied generative adversarial network (GAN) [31] to hyperspectral image classification. The method, that is based on GAN, does not need to provide the sample distribution in advance and the samples distribution can be learned automatically by the neural network. At the same time, the methods that are based on GAN are also able to use a huge number of unlabeled samples. 1DGAN and 3DGAN are used as spectral classification and spatial image block classification [32], respectively, by combining a conditional generative adversarial network (CGAN) [33] with an auxiliary classifier generative adversarial network (AC-GAN) [34], with category information as a conditional input. AC-GAN modifies its discriminator to a multi-category discriminator by viewing the generated samples as a new class and training the discriminator simultaneously with the real labeled samples. HSGAN [35] uses GAN to extract spectral features with discriminators in an unsupervised manner and train classifiers in a supervised manner, and finally the test samples labels are generated by voting on their neighboring image

element classes. Zhang M and Sun Q et al. used Wasserstein GANs (WGANs) [36] to extract features of hyperspectral image blocks by discriminators in an unsupervised manner, and then trained the classifier with a small number of samples. The above GAN-based methods utilize only spectral information or spatial information. Jie Feng [37] et al. combined the two and proposed a novel multi-class null-spectrum GAN method, which has two conditional generators to generate spectral samples and spatial image block samples of corresponding classes, and the discriminator takes both spectral and spatial image block samples as inputs, extracts their respective features, and then cascades them, and finally classifies them, and the generated samples are discriminated in the discriminator as not belonging to any class. These common GAN-based semi-supervised classification algorithms use discriminators to extract features for classification. Although the adversarial training of discriminators can enhance the ability of discriminators to extract features to a certain extent, the discriminators still need to force the distribution of the generated samples and real samples to be separated when the distributions are close to each other in the later stages of training, which deteriorates the classification ability of the discriminators. In general, the existing hyperspectral image classification algorithms that are based on GAN are not conducive to extracting discriminant information in the later training period because it uses discriminator as the feature extractor for classification. Although some methods expand the data by generating pseudo samples, they do not effectively use a large number of unlabeled samples to solve the problem of small samples.

In this paper, an adversarial representation learning that is based on generative adversarial networks (ARL-GAN) is proposed to extend GAN to the domain of representation learning. We used PCA to compress the raw data and added an encoder in front of the generator to combine it with the generator as a stack autoencoder. The discriminator is modified to be a multi-category discriminator and it is used instead of MSE to measure the difference between the generated image and the real image, so as to guide feature extraction and image generation through the high-level semantic information. In addition, conditional entropy is added to the objective function to increase the use of unlabeled samples in the network. This algorithm uses GAN to improve the representation capability of the encoder, and finally uses the features that are extracted by the encoder for classification, which is not affected by the above factors in the later stages of training, while training the GAN with a huge number of unlabeled samples. This makes the output features of the feature extractor are more conducive to classification, and also takes full advantage of the large amount of unlabeled data, which greatly improves the hyperspectral image classification performance under the small sample problem. The main contributions of this article are as follows.

1.  We construct a hyperspectral image block generator that is based on PCA, whose input is the feature vector that is extracted by the encoder rather than the noise, and the encoder is used as the feature extractor of classification.
2.  The distance of the class vector output by the discriminator replaces MSE in stacked autoencoder (SAE) to measure the error between the generated image block and the real image block, so that the features that are extracted by the encoder have more useful information for classification.
3.  GAN and conditional entropy are used to improve the utilization of unlabeled data when training discriminators, solving the problem of small samples in HSI classification.

The rest of this paper is organized as following. Section 2 describes the methodology of this paper in detail. Section 3 verifies the advancement of the proposed model by comparing it with other excellent HSI classification models. Finally, Section 4 summarizes this article.

## 2. Proposed Method

This section is structured as follows. First, we introduce the structure of the ARL-GAN algorithm that is proposed in this paper and its corresponding parameter settings. Second, we investigate the use of spectral regularization for ARL-GAN to stabilize the training

process. Third, we present the use of conditional entropy in detail. Finally, we describe the encoder *E*, the classifier *C*, the generator *G*, and the discriminator *D* of ARL-GAN.

### 2.1. ARL-GAN

The structure diagram of the ARL-GAN algorithm that is proposed in this paper is shown in Figure 1. ARL-GAN includes five parts. In the first part, PCA is used to extract the principal components in the spectral dimension of hyperspectral images so as to reduce redundancy and reduce the difficulty of image generation. In the second part, we extract the features of real hyperspectral image blocks using the encoder *E* based on a two-dimensional convolution neural network. In the third part, we generate (reconstruct) hyperspectral image blocks of the same class using the feature that is extracted by *E* as input using the generator *G* based on two-dimensional transposed convolution neural network. In the fourth part, we measure the generation errors of real hyperspectral image blocks and the generated hyperspectral image blocks by using a discriminator *D*, which is designed as a two-dimensional convolution neural network. The errors are calculated by the class similarity of the two image blocks. In the fifth part, we add a fully connected layer and the Softmax activation function after encoder *E* for classification.

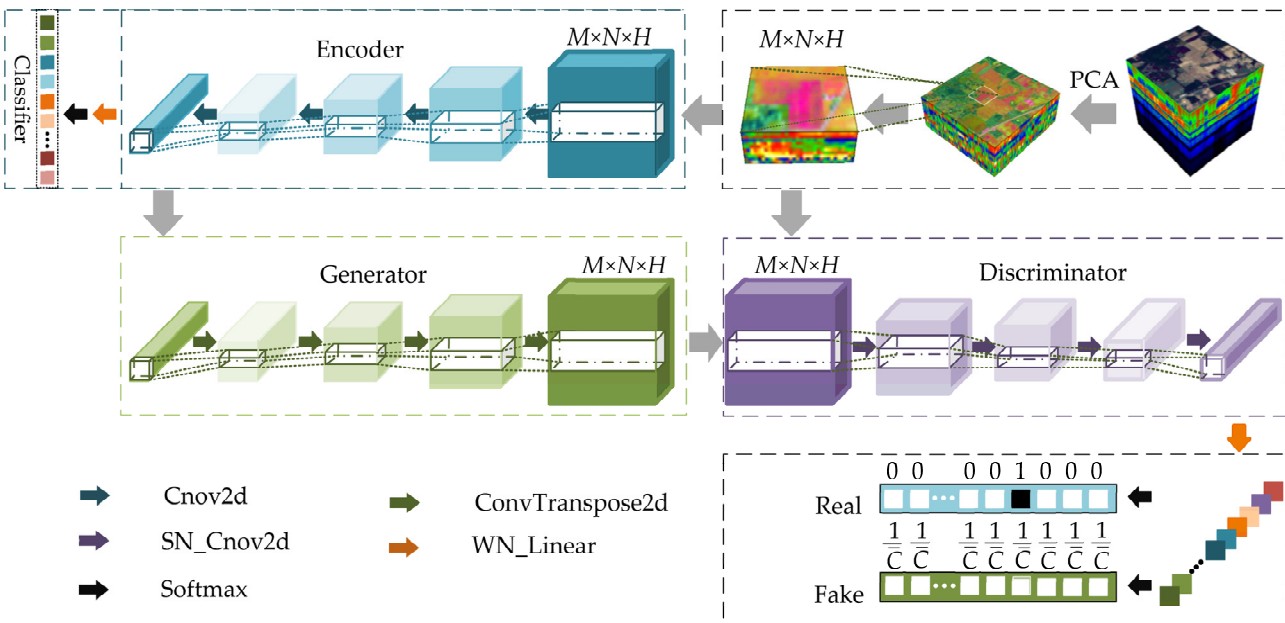

**Figure 1.** Structure diagram of the ARL-GAN algorithm.

In this paper, $x$ is used to represent the samples of HSI blocks with $M \times N \times H$ size, $M$ and $N$ are used to represent the length and width of the blocks, $H$ is used to indicate the number of channels of the HSI after PCA operation, and $y$ is used to represent one_hot label corresponding to the real sample $x$. The real labeled samples are represented as $(x_L, y_L)$, and the real unlabeled samples are represented as $x_U$. In the encoding phase, $x$ is used as input to the encoder $E$, and the encoded output is characterized by $f$. In the generation phase, the feature $f$ that is extracted by the encoder is used as input to the generator $G$, and the hyperspectral image blocks that are generated by the generator are represented as $x_{G(f)}$. The discriminator takes the resulting sample $x_{G(f)}$, the real labeled sample $(x_L, y_L)$, and the real untagged sample $x_U$ as input. The real labeled samples $(x_L, y_L)$ are used to learn the correct image block–category mapping function based on the discriminator. The unlabeled sample $x_U$ uses conditional entropy (CE) [38] to enhance discriminator classification, that is, to force the discriminator to output a class probability vector with an advantage class for the unlabeled sample. Conversely, the discriminator needs to output the class probability vectors with no dominant classes for $x_{G(f)}$ to distinguish between

the real samples and the generated samples. For generators, the dominant classes output by the discriminator are required to be consistent between the generated samples and the corresponding real samples. Finally, the feature that is extracted by the encoder is classified by a fully connected layer and Softmax score.

### 2.2. Spectral Regularization for Stabilizing ARL-GAN Training Process

GAN is prone to pattern collapse, gradient disappearance, and gradient explosion during the training process, leading to an extremely unstable training process. Therefore, many studies have been conducted to solve this problem. For example, WGAN and WGAN-GP solve this problem by modifying the loss function by replacing the Jensen-Shannon (JS) Divergence JS in the classical GAN loss function with the Wasserstein distance and implement the required Lipschitz continuity condition at that distance in two different ways. In addition to solving this problem by modifying the loss function, some researchers have proposed to solve this problem by modifying the network structure of generative adversarial networks. For example, the deep convolution generative adversarial network (DCGAN) [39], which mitigates the problem of network training instability through a well-designed network structure. In this paper, spectral normalization (SN) [40] is used to make the discriminator satisfy the Lipschitz continuity condition to solve the above problem.

SN applies regularization in the form of spectral norms over layer parameters, so that the discriminator $D$ has Lipschitz continuity conditions. Compared with the stable GAN approach of WGAN-GP, SN has a better effect in GAN [41]. The method that is presented in this paper requires the generation of HSI blocks, which is more difficult than the generation of hyperspectral vectors. Therefore, a spectral regularization method is used to stabilize the GAN.

In GAN, if the discriminator $D$ is K-Lipschitz continuous, any $x_1$ and $x_2$ in image space can be represented as Equation (1), where $\|\cdot\|_2$ represents the $L_2$ norm, $K$ is called the Lipschitz constant of $D(x)$, and the minimum of $K$ is called $\|D\|_{Lip}$. The spectral norm definition of the matrix is shown in the Formula (2), and $sup$ represents the upper bound.

$$\frac{\|D(x_1) - D(x_2)\|}{\|x_1 - x_2\|_2} \leq K, \forall x_1, x_2 \tag{1}$$

$$\|A\|_2 = sup \frac{\|Ax\|_2}{\|x\|_2} = \sigma(A) \tag{2}$$

To analyze whether the discriminator $D$ satisfies the Lipschitz continuity condition after $SN$ is used, the discriminator function can be expressed as $x^l = a^l(W^l x^{l-1} + b^l)$, if $D$ is a fully connected network with a non-linear activation function, each layer is recorded as $\theta = \left\{ W^l, b^l \right\}_{l=1}^{L}$, and the parameter of $D$ is recorded as $D_\theta(x^0) = x^L$. Within a small neighborhood of the value, the discriminator can be considered as a linear function, $D_\theta(x) = W_{\theta,x} x + b_{\theta,x}$. Formula (3) can be derived from the property of matrix norm $\sigma(AB) \leq \sigma(A)\sigma(B)$.

The activation function uses ReLU, and $a^l(x^{l-1})$ can be thought of as $Diag_{\theta,x}^l x^{l-1}$. $Diag_{\theta,x}^l$ represents a diagonal matrix, with 1 at the non-negative corresponding position of $x^{l-1}$ and 0 at the rest, so that there is $\sigma(Diag_{\theta,x}^l) = \|Diag_{\theta,x}^l\|_2 \leq 1$, from which the Formulas (4) and (5) can be derived.

$$\frac{\|D_\theta(x+\delta) - D_\theta(x)\|_2}{\|\delta\|_2} = \frac{\|W_{\theta,x}\delta\|_2}{\|\delta\|_2} \leq \sigma(W_{\theta,x}) = \sup_{\delta \neq 0} \frac{\|W_{\theta,x}\delta\|_2}{\|\delta\|_2} \tag{3}$$

$$W_{\theta,x} = \prod_{l=1}^{L} Diag_{\theta,x}^l W_{\theta,x}^l \tag{4}$$

$$\sigma(W_{\theta,x}) \leq \prod_{l=1}^{L} \sigma(Diag_{\theta,x}^l)\sigma(W_{\theta,x}^l) \leq \prod_{l=1}^{L} \sigma(W^l) \tag{5}$$

If SN is used for each network layer of discriminator $D$, the Formulas (6)–(8) can be derived, and then discriminator $D$ satisfies the 1-Lipschitz continuous condition.

$$\hat{W}_{SN}^{l} = \frac{W^{l}}{\sigma(W^{l})} \qquad (6)$$

$$\sigma(\hat{W}_{SN}^{l}) = \frac{\sigma(W^{l})}{\sigma(W^{l})} = 1 \qquad (7)$$

$$\frac{\|D_{\theta}(x + \delta) - D_{\theta}(x)\|_{2}}{\|\delta\|_{2}} \leq \sigma(W_{\theta,x}) \leq 1 \qquad (8)$$

SN is essentially a layer-by-layer singular value decomposition (SVD) process by dividing the parameter matrix of each layer by the spectral norm of the parameter matrix, but the process of solving SVD is time-consuming, so it is approximated by power iteration. The entire algorithm flow is shown in Algorithm 1.

| **Algorithm 1.** Spectral Normalization (SN) | |
|---|---|
| Step1: | For $l = 1, \cdots, L$ network layers, the random initialization vector $\widetilde{u}_{l}$ with Gaussian distribution is used; The number of power iterations is $N$. |
| Step2: | Calculate the values of $\widetilde{v}_{l}$ and $\widetilde{u}_{l}$:<br>For $l = 1, \cdots, L$<br>For $n_{1} = 1, \cdots, N_{1}$<br>$\widetilde{v}_{l} = (W^{l})^{T}\widetilde{u}_{l} / \|(W^{l})^{T}\widetilde{u}\|_{2}$<br>$\widetilde{u}_{l} = W^{l}\widetilde{v}_{l} / \|W^{l}\widetilde{v}_{l}\|_{2}$ |
| Step3: | Calculate spectral norm normalization parameters according to $\widetilde{v}_{l}$ and $\widetilde{u}_{l}$: $\overline{W}_{SN}^{l}(W^{l}) = W^{l}/\sigma(W^{l})$, where $\sigma(W^{l}) = \widetilde{u}_{l}^{T}W^{l}\widetilde{v}_{l}$. |
| Step4: | On the training dataset $D_{M} = \{(x_{i_{1}}, y_{i_{1}}), \cdots, (x_{i_{k}}, y_{i_{k}})\}$ of mini-batch, update the parameter $W^{l}$ with the learning rate $\alpha$:<br>$W^{l} = W^{l} - \alpha\nabla_{W^{l}}\ell(\overline{W}_{SN}^{l}(W^{l}), D_{M})$ |

*2.3. Conditional Entropy*

The cost of labeled samples in HSI datasets is high and the available labeled samples are limited, so the introduction of unlabeled samples into the deep learning algorithm needs to be considered to enhance the classification ability of the model. Classical GAN is an unsupervised training method, but the generated samples have a poor enhancement effect on the classifier, so both CGAN and ACGAN introduce category information as additional information to constrain GAN. The algorithm in this paper also introduces category information by changing the discriminator to a multi-classification discriminator. For unlabeled samples without category information, the algorithm in this paper exploits it by adding conditional entropy to the objective function.

To enhance the ability of the generator to generate samples, the discriminator needs to have a strong ability to discriminate between true and false for unlabeled data. However, the binary classification objective function of the classical GAN does not force a dominant class, and only needs to satisfy Equation (9) to obtain the decision boundary of the correct data source, while the class probability $P_{D}(y_{c}|x)$ of the real unlabeled samples may be evenly distributed. In order to ensure the discriminator has strong ability to distinguish true from false, a conditional entropy is added to the algorithm in this paper. At the same time, in practice, the real labeled samples do not know the specific label, but they should belong to a class, so conditional entropy is added as an a priori condition in the classifier to enhance the performance of the classifier. The mathematical expression of conditional entropy is shown in Equation (10). When the probability of one class is 1 and the predicted probability of other classes is 0, the conditional entropy obtains the maximum value and

when the predicted probability of each class is equal, the conditional entropy obtains the minimum value.

$$\sum_{c=1}^{\overline{C}} P_D(y_c|x) > P_D(y_{\overline{C}+1}|x) \tag{9}$$

$$CE = E_{x_U} \sum_{c=1}^{\overline{C}} p_C(y_c|x) \log p_C(y_c|x) \tag{10}$$

### 2.4. Encoder E and Classifier C of ARL-GAN

The role of the encoder $E$ is to map the input hyperspectral image block $x_{L\cup U}$ from the input space to the feature space $f$. The encoder maps the hyperspectral image block with original size of $M \times N \times H$ to a feature vector with size of $1 \times 1 \times F$ by four-step convolution. ReLU is used as the nonlinear activation function for each layer. The process of extracting the features by the encoder can be represented as Equation (11).

$$f = E(x_{L\cup U}) \tag{11}$$

The general semi-supervised classification algorithm that is based on GAN uses a discriminator to extract features for classification. Although the adversarial training of the discriminator can enhance the ability of the discriminator to extract features to a certain extent, the discriminator still needs to force the distribution of the generated samples and the real samples to be separated when the distribution is close to each other in the later stage of training, which affects the classification ability of the discriminator. In contrast, the algorithm in this paper uses GAN to improve the representation capability of the encoder, and ultimately uses the features that are extracted by the encoder for classification, which is not affected by the above factors in the later stages of training. Compared with the semi-supervised SAE algorithm without the antagonistic training, the feature that is extracted by the encoder in this paper is more suitable for classification tasks. The classifier adds a full connection layer with weight normalization (WN) [42] to the encoder $E$. The input image block's class probability distribution $p_C(y_c|x)$ is obtained by passing the output of the full connection layer through the Softmax, as shown in Formula (12). The objective function of classifier $C$ can be expressed as Formula (13).

$$p_C(y_c|x) = \text{softmax}\,(Wn\_Fc(f)) \tag{12}$$

$$\max L_C = \text{E}_{(x_L, y_L)} \log p_C(y_c|x, y) + \lambda \text{E}_{x_U} \sum_{c=1}^{\overline{C}} p_C(y_c|x) \log p_C(y_c|x) \tag{13}$$

In (13) the first term represents the cross-entropy that is calculated from the labeled samples, the second term represents the conditional entropy that is calculated from the unlabeled samples, and $\lambda$ is the weight of the conditional entropy. As whether the predicted result of the classifier is correct or not, the conditional entropy will promote the network to further enhance the current prediction. If the classifier is immature and has poor recognition ability, it will lead to the degradation of classification. Therefore, when the number of iterations is low in the early stages of training, $\lambda$ is small. When the classifier is fully trained with the increase of training steps, the accuracy of the prediction results increases, and the weight $\lambda$ should also increase with the increase of steps.

### 2.5. Generator G and Discriminator D of ARL-GAN

The generator consists of four transposed convolution layers, which take the feature that is extracted by the encoder with a size of $1 \times 1 \times F$ as input to generate (rebuild) an image block with a size of $M \times N \times H$, using ReLU as the activation function and Tanh as the activation function in the output layer to match the range of the true hyperspectral data. The discriminator is designed by ten layers of convolution layers that are normalized by spectral norms and full connection layer normalized by weights. Because the discriminator

is not only used to correctly classify the real labeled samples, but also to determine which is the generated sample and which is the real sample, the task is heavier than the classifier, so more network layers and more network parameters are used. The discriminator uses LeakyReLU as the activation function except for the final fully connected layer, which uses the Softmax function. The arrows represent the direction of data flow.

Generator $G$ is used to generate hyperspectral image blocks using the features that are extracted by the encoder $E$ as input. The stacked autoencoder uses the root mean square error to calculate the reconstruction error so that the input image block and the reconstructed image block are identical at the pixel level. The encoder of SAE can be considered as a non-linear dimension reduction algorithm. Although the redundancy of data is reduced, the ability of the encoder to extract features is improved to a certain extent by. However, the features that are extracted by this encoder are not completely suitable for classification tasks. In this algorithm, MSE is replaced by a discriminator, which takes the generated image block and the real image block as input, and then constrains of the generator according to the advanced semantic information of category, making the generated image block consistent with the corresponding real image block category, thus making the features and categories extracted by the encoder highly relevant. The extracted features contain the main information of the class and remove the remaining noise information. The encoder and generator are trained with a discriminator to further enhance the ability of encoder and generator.

The generator's target function is shown in Equation (14), where $p_D(y_c|x)$ represents the predictive probability of the discriminator, $G(f)$ represents the generated data, $p_D(y_c|x_{L\cup U})$ denotes the class probability of the discriminator for the sample output, and $p_D(y_c|x_{G(f)})$ denotes the class probability of the discriminator for the generated sample output. This objective function can constrain the generation process so that the generated image block and the real image block belongs to the same class, rather than making them identical at the pixel level. The generators and encoders are a complete generation network when updating parameters, so the parameters of encoders $E$ and $G$ are updated during reverse propagation.

$$\min L_{G,E} = \mathrm{E}_x \left( \frac{1}{\overline{C}} \sum_{c=1}^{\overline{C}} \left( p_D\left(y_c\Big|x_{L\cup U}\right) - p_D\left(y_c\Big|x_{G(f)}\right) \right)^2 \right) \tag{14}$$

The function of the discriminator is mainly to distinguish the generated image blocks from the real data blocks. However, unlike the classical GAN which uses binary classification to discriminate, the discriminator $D$ in the proposed method is a multi-class discriminator. The objective function of $D$ is shown in Equation (15). The objective function of the discriminator contains three terms, the first term is the cross-entropy for the true labeled samples; the second term is the conditional entropy for the real labeled samples, which gives the discriminator an advantage class for the class probability of the output of the real labeled samples; and the third term is the objective function for generating image blocks. Unlike other semi-supervised GAN algorithms, the generated samples are considered as a new class [43,44]. The label of the sample that is generated by this algorithm is $y_G = [1/\overline{C}, \cdots 1/\overline{C}]$ where $\overline{C}$ is the number of classes, which is also contrary to the true unlabeled sample, which does not have any dominant classes and outputs a uniform class probability distribution.

$$\max L_D = E_{(x_L,y_L)} \log p_D(y_c|x,y) + \lambda E_{x_U} \sum_{c=1}^{\overline{C}} p_D(y_c|x) \log p_D(y_c|x) + E_{x_{G(f)}} \sum_{c=1}^{\overline{C}} y_G(c) \log(p_D(y_c|x)) \tag{15}$$

The optimization process of encoder $E$, classifier $C$, generator $G$, and discriminator $D$ in the whole network uses an alternating optimization strategy. When updating the parameters, $G$ and $E$ are considered as a whole generation network. When $C$ and $G(E)$ network parameters are fixed, $D$'s network parameters are optimized. Conversely, when

*D* is frozen, the parameters of the remaining networks are updated. All the networks are optimized using the Adam optimization algorithm. Through this end-to-end network structure and alternate optimization, the test samples are finally classified by classifier *C*.

## 3. Experiments and Result Analysis

In this section, three well-known HSI datasets are described, then we introduce the implementation details of the algorithm in this paper and compare it with other excellent methods. The effectiveness of adversarial learning and conditional entropy are analyzed experimentally and a sensitivity analysis with the number of labeled samples is performed. Finally, a comparison with existing GAN-based hyperspectral image classification algorithms is performed to evaluate the HSI image classification performance of the proposed algorithm.

### 3.1. Hyperspectral Datasets

In this experiment, three widely-used datasets were used and they are Indian Pines dataset, Pavia University dataset (PaviaU), and Salinas. Figure 2a–c shows the false-color image and ground truth of Indian Pines, Pavia University, and Salinas dataset, respectively.

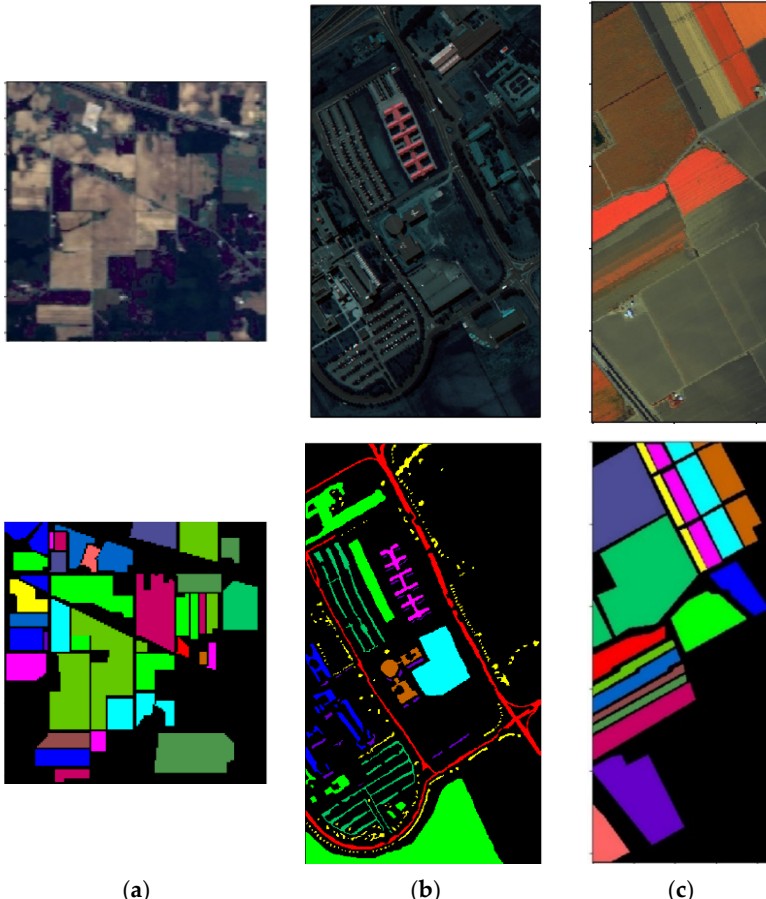

(**a**)  (**b**)  (**c**)

**Figure 2.** False-color composites (first row) and ground truths (second row) of experimental HSI datasets. Each color represents one kind of object. (**a**) Indian Pines; (**b**) PaviaU; (**c**) Salinas.

The Indian Pines dataset was gathered by the AVIRIS sensor in the northeast of Indiana. It is 145 × 145 in size and has a spatial resolution of 20 m per pixel. A total of 20 bands were removed due to water absorption, leaving 200 bands for our experimental analysis, with a spectral range of 400 to 2500 nm. This dataset contains 16 categories and background information.

The PaviaU dataset was gathered in northern Italy by the Reflective Optics System Imaging Spectrometer. Its spatial size is 610 × 340 and the resolution is 1.3 m per pixel. It has 103 bands, which range from 430 to 860 nm.

The Salinas dataset was gathered by the AVIRIS sensor over the Salinas Valley. The dataset is of size 512 × 217 × 224, and it has a spatial resolution of 3.7 m per pixel with 16 land-cover classes.

Real samples were divided into a training set and a test set by random sampling. A total of 5% samples were selected in Indian Pines, 1% samples were selected from the Salina dataset, and 3% samples were selected from the PaviaU dataset. The values of all HSIs datasets are normalized between −1 and 1, and then the spectral dimension is reduced to 10 dimensions by the PCA algorithm. The size of the input image block is 32 × 32 × 10 in Indian Pines and Salinas, and the size of the input image block is 8 × 8 × 10 in PaviaU, because PaviaU is a collection of hyperspectral image data for urban areas, while Indian Pines and Salinas are a hyperspectral dataset for agricultural areas, where the distribution of land objects is more complex and homogeneous. Therefore, the size of the input image block that was selected by this algorithm on the PaviaU dataset is small. In the training, the number of training samples per small batch is set to 32; the learning rate is set by simulated annealing algorithm, the learning rate range is 0.0 to 0.002; the conditional entropy weight $\lambda$ uses step parameters, the range is 0.5 to 1, and the $\lambda$ increases by 0.05; and the total number of training steps is 1000 for each 100 step increase in training steps. The experiments in this paper were run 10 times independently, and each time the training set and test set were randomly divided, and the average and standard deviation of the experimental results were calculated. All the experiments in this paper are run on an Nvidia 1080ti graphics card. Moreover, Tables 1–3 listed the land cover type and the total number of samples in the three HSI datasets.

**Table 1.** Land cover types and total number of samples in the Indian Pines dataset.

| No. | Color | Name | Number | No. | Color | Name | Number |
|-----|-------|------|--------|-----|-------|------|--------|
| 1 | | Alfalfa | 46 | 9 | | Oats | 20 |
| 2 | | Corn-notill | 1428 | 10 | | Soybean-notill | 972 |
| 3 | | Corn-mintill | 830 | 11 | | Soybean-mintill | 2455 |
| 4 | | Corn | 237 | 12 | | Soybean-clean | 593 |
| 5 | | Grass-pasture | 483 | 13 | | Wheat | 205 |
| 6 | | Grass-trees | 730 | 14 | | Woods | 1265 |
| 7 | | Grass-pasture-mowed | 28 | 15 | | Buildings-Grass-Trees | 386 |
| 8 | | Hay-windrowed | 478 | 16 | | Stone-Steel-Towers | 93 |
| | | Total Numbers | | | | 10,249 | |

**Table 2.** Land cover types and total number of samples in the PaviaU dataset.

| No. | Color | Name | Number | No. | Color | Name | Number |
|-----|-------|------|--------|-----|-------|------|--------|
| 1 | | Asphalt | 6631 | 6 | | Bare Soil | 5029 |
| 2 | | Meadows | 18,649 | 7 | | Bitumen | 1330 |
| 3 | | Gravel | 2099 | 8 | | Self-Blocking Bricks | 3682 |
| 4 | | Trees | 3064 | 9 | | Shadows | 947 |
| 5 | | Painted metal sheets | 1345 | | | Total Numbers | 42,776 |

**Table 3.** Land cover types and total number of samples in the Salinas dataset.

| No. | Color | Name | Number | No. | Color | Name | Number |
|---|---|---|---|---|---|---|---|
| 1 | | Brocoli_green_weeds_1 | 1977 | 9 | | Soil_vinyard_develop | 6197 |
| 2 | | Brocoli_green_weeds_2 | 3726 | 10 | | Corn_senesced_green_weeds | 3249 |
| 3 | | Fallow | 1976 | 11 | | Lettuce_romaine_4wk | 1058 |
| 4 | | Fallow_rough_pow | 1394 | 12 | | Lettuce_romaine_5wk | 1908 |
| 5 | | Fallow_ smooth | 2678 | 13 | | Lettuce_romaine_6wk | 909 |
| 6 | | Stubble | 3959 | 14 | | Lettuce_romaine_7wk | 1061 |
| 7 | | Celery | 3579 | 15 | | Vinyard_untrained | 7164 |
| 8 | | Grapes_untrained | 11,213 | 16 | | Vinyard_vertical_trellis | 1737 |
| | | Total Numbers | | | | 53,785 | |

In order to quantitatively evaluate the experimental results, three common evaluation criteria are used: OA, AA, and Kappa coefficients. The definitions of all evaluation criteria are shown as follows:

(1) OA: OA assesses the proportion of correctly identified samples to all the samples.

$$\text{OA} = \frac{\sum\limits_{i=1}^{\overline{C}} h_{ii}}{N} \tag{16}$$

where $N$ is the total number of labeled samples, $h_{ii}$ represents the number of class $i$ samples divided into class $i$, and $\overline{C}$ is the total number of categories.

(2) AA: AA represents the mean of the percentage of the correctly identified samples.

$$\text{AA} = \frac{1}{\overline{C}} \sum\limits_{i=1}^{\overline{C}} \frac{h_{ii}}{N_i} \tag{17}$$

where $\overline{C}$ is the total number of categories, $h_{ii}$ represents the number of samples of category $i$ divided into category $i$, and $N_i$ represents the number of samples of category $i$.

(3) Kappa: Kappa coefficient denotes the interrater reliability for categorical variables.

$$\text{Kappa} = \frac{N\sum\limits_{i}^{\overline{C}} h_{ii} - \sum\limits_{i}^{\overline{C}} (h_{i+} \cdot h_{+i})}{N^2 - \sum\limits_{i}^{\overline{C}} (h_{i+} \cdot h_{+i})} \tag{18}$$

where $h_{i+}$ and $h_{+i}$, respectively, represent the total number of samples of category $i$ true category and the number of samples predicted to be category $i$.

### 3.2. Visual Analysis of Generated Image Blocks

In order to display the generated image blocks conveniently, the original dataset is first reduced to 3-dimensional using PCA, and the last output channel of the generated network is changed from 10 to 3 to generate three-channel image blocks as shown in Figure 3.

The real image blocks that are drawn with pseudo color images and the corresponding generated image blocks are also provided. By comparing the real image block and the generated image block, we can see that they are not exactly the same. The generated image block and the real image block are very similar in the central pixel part, but the surrounding pixels are not exactly the same because the generation model of the algorithm in this paper only makes the two categories consistent, not on the pixels, so this phenomenon meets the theoretical requirements of the algorithm.

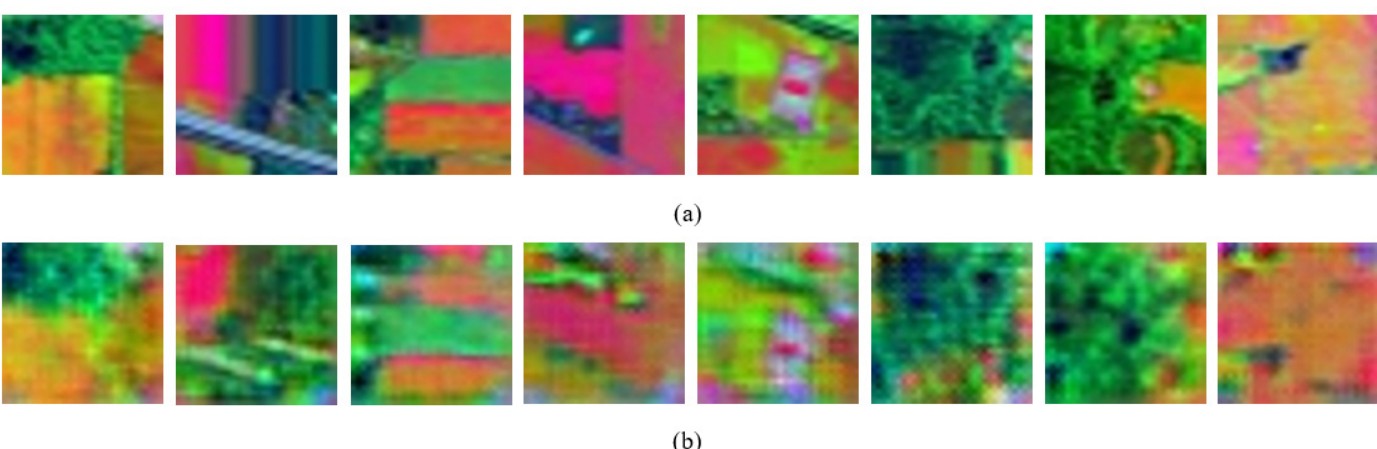

**Figure 3.** Visualization of the real image blocks (**a**) and the corresponding resulting image block (**b**).

### 3.3. Numerical Results and Visualization of HSI Classification

To evaluate the performance of this classification algorithm, this section compares it with other algorithms. In this paper, six representative hyperspectral image classification algorithms are selected, which are RBF-SVM [9], SAE [17], DBN [45], CNN [46], PPFCNN [26], and 3DCNN [47]. RBF-SVM is a classical SVM method that is based on radial basis function and the other methods are based on deep learning algorithm. The SVM in the RBF-SVM algorithm extends it from a two-class model to a multi-class model through a one-to-many strategy. The window size and other parameter settings in SAE, DBN, CNN, PPFCNN, and 3DCNN are the optimal experimental settings that are directly used in the original papers. The spatial dimensions of PPFCNN sample pairs are also set according to the original paper. The window size in a 3DCNN is determined by searching for the optimal size within a range of 5 to 33.

The classification results mainly include two parts, one are the numerical classification results, the other is the visual classification results. These two classification results on the Indian Pines, PaviaU, and Salinas datasets are shown, respectively.

Indian Pines: The numerical classification results for the Indian Pines dataset are shown in Table 4. The first three lines show AA, OA, and Kappa coefficients that were calculated from all samples. The last 16 rows record the classification accuracy of the each classes. Under the three evaluation criteria of AA, OA, and Kappa coefficients, the proposed algorithm performs better than the other six algorithms. Compared with RBF-SVM, the classification accuracy of the other five classification algorithms based deep learning has been greatly improved, which shows that the deep learning algorithm can extract distinguished features more effectively. PPF-CNN is more effective than CNN in this dataset because the number of training samples is increased and the classification accuracy of the model is improved by using pairs of samples. Compared with SAE, this algorithm increases the OA, AA, and Kappa coefficients by 16.4%, 25.6%, and 18.7%, respectively. This shows that the features that are extracted by the encoder in this algorithm are more suitable for classification. Three-dimensional convolution can effectively use the spatial-spectral information to extract the combined features of spatial spectrum and improve the classification accuracy. Compared with three-dimensional CNN, the algorithm in this paper improves the OA, AA, and Kappa coefficients by 5.5%, 5.6%, and 6.1%, respectively. The optimal classification results for each class are represented by bold numbers in the table. There are 13 classes (16 classes in total) of the algorithm that have obtained the optimal classification results. At the same time, it can be found that RBF-SVM and PF-CNN have zero classification accuracy for the seventh and ninth classes because the number of samples is much less than other classes. The accuracy of the algorithms in this paper is $82.9 \pm 16.2$ and $92.0 \pm 16.0$, respectively. At the same time, the proposed method achieves

the optimal value on AA. Therefore, the algorithm in this paper can effectively handle the extreme sample imbalance problem on the Indian Pines dataset and achieve good results.

**Table 4.** Classification comparison result of ARL-GAN with different algorithms on Indian Pines dataset.

| Method | RBF-SVM | DBN | SAE | CNN | 3DCNN | PPF-CNN | ARL-GAN |
|---|---|---|---|---|---|---|---|
| OA (%) | 77.8 ± 0.8 | 80.6 ± 0.1 | 81.9 ± 0.1 | 85.4 ± 0.8 | 92.8 ± 0.8 | 87.9 ± 0.8 | **98.3 ± 0.4** |
| AA (%) | 61.3 ± 1.4 | 68.3 ± 1.7 | 69.4 ± 1.9 | 79.4 ± 1.6 | 89.4 ± 1.4 | 76.5 ± 0.6 | **95.0 ± 2.1** |
| Kappa | 74.5 ± 1.0 | 77.8 ± 1.3 | 79.3 ± 1.1 | 84.3 ± 2.9 | 91.9 ± 0.9 | 86.3 ± 0.9 | **98.0 ± 0.4** |
| 1 | 6.1 ± 11.2 | 13.6 ± 5.6 | 10.0 ± 6.4 | 78.4 ± 10.2 | 50.4 ± 8.4 | 50.4 ± 8.4 | **80.4 ± 26.2** |
| 2 | 72.9 ± 3.6 | 79.8 ± 2.9 | 79.7 ± 2.3 | 75.4 ± 2.4 | 92.7 ± 3.5 | 89.2 ± 2.1 | **99.0 ± 1.0** |
| 3 | 58.0 ± 3.6 | 70.5 ± 2.2 | 74.9 ± 4.8 | 82.8 ± 3.3 | 87.2 ± 10.4 | 77.1 ± 2.7 | **96.7 ± 1.8** |
| 4 | 39.0 ± 15.0 | 71.3 ± 6.6 | 62.8 ± 8.3 | 89.2 ± 3.5 | 83.4 ± 8.3 | 87.7 ± 3.7 | **97.5 ± 4.9** |
| 5 | 87.0 ± 4.5 | 80.1 ± 4.1 | 84.2 ± 3.3 | 69.0 ± 4.6 | 84.0 ± 5.7 | 92.7 ± 1.0 | **96.3 ± 1.1** |
| 6 | 92.4 ± 2.0 | 94.2 ± 2.4 | 94.3 ± 1.7 | 92.8 ± 2.5 | 93.4 ± 2.5 | 93.1 ± 1.9 | **99.2 ± 1.0** |
| 7 | 0 ± 0 | 28.1 ± 22.6 | 24.4 ± 18.8 | 51.1 ± 12.3 | **97.2 ± 4.8** | 0 ± 0 | 82.9 ± 16.2 |
| 8 | 98.1 ± 1.4 | 98.5 ± 1.5 | 98.8 ± 0.4 | 97.1 ± 1.6 | 97.4 ± 2.8 | 99.6 ± 0.3 | **100.0 ± 0.0** |
| 9 | 0 ± 0 | 9.5 ± 2.4 | 11.1 ± 10.1 | 41.6 ± 9.9 | 77.0 ± 11.1 | 0 ± 0 | **92.0 ± 16.0** |
| 10 | 65.8 ± 3.7 | 73.2 ± 4.7 | 73.6 ± 3.8 | 81.0 ± 2.6 | 93.3 ± 5.0 | 85.6 ± 2.8 | **96.2 ± 3.1** |
| 11 | 85.3 ± 2.9 | 82.7 ± 2.2 | 83.4 ± 2.0 | 87.2 ± 1.5 | 94.9 ± 2.7 | 83.8 ± 1.6 | **99.3 ± 0.7** |
| 12 | 69.6 ± 6.5 | 62.0 ± 5.8 | 70.4 ± 8.0 | 84.4 ± 2.3 | 89.8 ± 4.3 | 90.4 ± 3.1 | **97.3 ± 3.2** |
| 13 | 92.3 ± 4.1 | 89.7 ± 10.6 | 94.2 ± 4.3 | 83.1 ± 4.2 | 92.8 ± 5.9 | **97.8 ± 0.9** | 96.9 ± 3.8 |
| 14 | 96.6 ± 1.0 | 94.4 ± 1.6 | 94.2 ± 1.5 | 98.2 ± 0.8 | 98.3 ± 1.3 | 95.5 ± 1.1 | **100.0 ± 0.0** |
| 15 | 41.7 ± 7.0 | 64.2 ± 6.5 | 66.1 ± 5.6 | 84.7 ± 4.5 | 77.8 ± 13.4 | 78.0 ± 2.4 | **99.1 ± 1.7** |
| 16 | 75.2 ± 9.0 | 80.5 ± 13.2 | 87.6 ± 8.1 | 76.0 ± 8.1 | 88.4 ± 5.3 | **97.3 ± 1.3** | 88.0 ± 3.0 |

The bold value indicates the optimal value.

Figure 4 shows the visual classification results of the different algorithms on the Indian Pines dataset. First, we can see that the RBF-SVM, DBN, and other spectral classification algorithms only use spectrum to classify. There are a lot of noise points in the classification results, while the 3DCNN and the proposed method both use spatial information and have relatively few noise points. The classification map of the 3DCNN contains a large number of error points at the boundary. The algorithm in this paper has fewer error points than that of 3DCNN for those pixels that are close to the boundary. It is concluded that the algorithm that is presented in this paper can reduce the noise points and maintain the classification boundary better than other algorithms.

PaviaU: The classification results of the PaviaU dataset are shown in Table 5. Under the three evaluation criteria of AA, OA, and Kappa coefficient, the classification effect of the algorithm that is proposed in this paper is better than the other six algorithms. Consistent with the conclusion for the Indian dataset, the classification effect of the six classification algorithms that are based on deep learning are better than the traditional algorithm that is based on SVM. The classification effect of 3DCNN on PaviaU is lower than that of PPF-CNN, which is different from the conclusion on Indian Pines dataset. It can be seen that these two algorithms are more sensitive to datasets. The algorithm in this paper has achieved the best classification effect on these two datasets as it shows it is less sensitive to datasets than other algorithms and is easy to be transplanted to different datasets for classification. Compared with the SAE algorithm on this dataset, the OA, AA, and Kappa coefficients of the proposed ARL-GAN have increased by 7.4%, 10.3%, and 9.8%, respectively. Since the classification accuracy of all algorithms on PaviaU is higher than that for the Indian Pines dataset, the growth rate on this dataset is lower than that for the Indian Pines dataset, However, it can also be verified that the features that were extracted by the encoder through the algorithm in this paper are more suitable for classification. Among the six comparison algorithms, PPF-CNN achieved the best classification results on the dataset. Compared with PPF-CNN, the OA, AA, and Kappa coefficients of this algorithm were improved by 2.9%, 4.5%, and 3.7%, respectively. At the same time, our algorithm achieves the best classification accuracy on all classes. Figure 5 shows the visual

classification results of different algorithms on the PaviaU dataset, from which we can obtain the similar conclusion for the Indian Pines dataset. The visual classification results of the algorithm in this paper have less noise points, and shows better performance for those boundary samples.

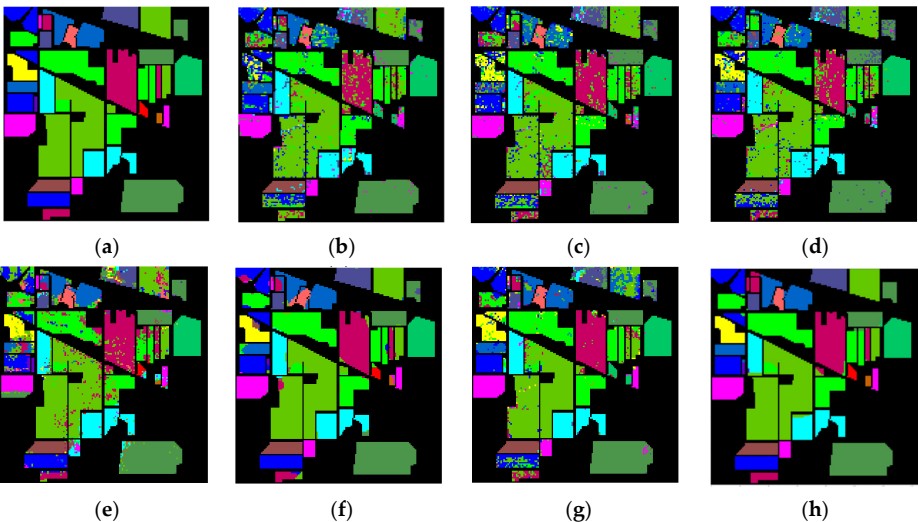

**Figure 4.** Visualization of the classification results on the Indian Pines dataset. (**a**) Ground Truth. (**b**) RBF-SVM. (**c**) DBN. (**d**) SAE. (**e**) CNN. (**f**) 3DCNN. (**g**) PPF-CNN. (**h**) ARL-GAN.

**Table 5.** Classification comparison results of ARL-GAN with different algorithms on the PaviaU dataset.

| Method | RBF-SVM | DBN | SAE | CNN | 3DCNN | PPF-CNN | ARL-GAN |
|---|---|---|---|---|---|---|---|
| OA (%) | 90.3 ± 0.6 | 91.9 ± 0.3 | 92.4 ± 0.3 | 93.0 ± 0.6 | 95.2 ± 0.7 | 96.9 ± 0.2 | **99.8 ± 0.1** |
| AA (%) | 86.6 ± 0.9 | 88.5 ± 0.8 | 89.3 ± 0.7 | 88.6 ± 0.8 | 91.2 ± 1.1 | 95.1 ± 0.2 | **99.6 ± 0.2** |
| Kappa | 87.1 ± 0.8 | 89.2 ± 0.4 | 89.9 ± 0.4 | 90.7 ± 0.7 | 93.8 ± 0.9 | 96.0 ± 0.2 | **99.7 ± 0.1** |
| 1 | 90.7 ± 1.1 | 91.6 ± 0.8 | 92.3 ± 1.1 | 93.1 ± 1.4 | 95.5 ± 1.2 | 98.0 ± 0.1 | **100.0 ± 0.0** |
| 2 | 96.8 ± 0.7 | 97.4 ± 0.4 | 97.6 ± 0.3 | 97.6 ± 0.9 | 99.4 ± 0.3 | 99.2 ± 0.2 | **100.0 ± 0.0** |
| 3 | 60.2 ± 5.4 | 69.7 ± 6.0 | 72.1 ± 3.5 | 77.9 ± 4.5 | 92.6 ± 5.4 | 84.9 ± 1.8 | **97.9 ± 1.4** |
| 4 | 90.8 ± 2.0 | 91.2 ± 1.4 | 90.9 ± 1.4 | 86.4 ± 3.6 | 75.2 ± 4.9 | 95.8 ± 0.8 | **99.0 ± 0.6** |
| 5 | 98.8 ± 0.4 | 98.6 ± 0.6 | 98.7 ± 0.4 | 98.5 ± 1.4 | 95.4 ± 4.3 | 99.8 ± 0.1 | **100.0 ± 0.0** |
| 6 | 79.5 ± 4.9 | 85.6 ± 2.2 | 86.9 ± 1.9 | 91.0 ± 2.8 | 99.4 ± 0.6 | 96.4 ± 0.3 | **100.0 ± 0.0** |
| 7 | 74.3 ± 5.1 | 74.8 ± 4.8 | 78.1 ± 4.9 | 81.2 ± 2.9 | 91.5 ± 3.4 | 89.2 ± 0.8 | **100.0 ± 0.0** |
| 8 | 88.8 ± 2.2 | 88.2 ± 1.3 | 87.8 ± 1.4 | 92.5 ± 2.2 | 94.8 ± 1.4 | 93.7 ± 1.2 | **100.0 ± 0.0** |
| 9 | 99.8 ± 0.1 | 99.6 ± 0.1 | 99.5 ± 0.3 | 79.0 ± 4.1 | 77.4 ± 2.8 | 98.5 ± 0.7 | **100.0 ± 0.0** |

The bold value indicates the optimal value.

Salinas: The classification results of the Salinas dataset are shown in Table 6. Firstly, from the three evaluation criteria of AA, OA, and Kappa coefficients, the classification effect of the algorithm that is proposed in this paper is better than the other six algorithms. Consistent with the conclusions from the the first two datasets, the classification effect of the six classification algorithms that are based on deep learning is better than the traditional methods based on SVM. On Salinas, the classification effect of 3DCNN is higher than that of PPF-CNN, which is consistent with the conclusion for the Indian Pines dataset. Therefore, based on the results of the three datasets, the classification result of 3DCNN is better than that of PPF-CNN as a whole thanks to its comprehensive utilization of spatial-spectral information by means of three-dimensional convolution. Compared with the SAE algorithm, the algorithm in this paper increases the OA, AA, and Kappa coefficients by 8.4%, 5.0%, and 9.3%, respectively. For the three datasets, the algorithm in this paper has a greater improvement than the SAE algorithm in the three evaluation index values, so it further proves that the features that are extracted by the encoder of this paper algorithm

are more suitable for classification. Among the six algorithms, 3DCNN achieves the best classification results on the dataset. Compared with 3DCNN, the proposed method improves the OA, AA, and Kappa coefficients by 2.0%, 1.9%, and 2.0%, respectively. The algorithm in this paper achieves the optimal classification accuracy in all classes, with 100% accuracy in 10 classes and nearly 100% accuracy in the other 6 classes.

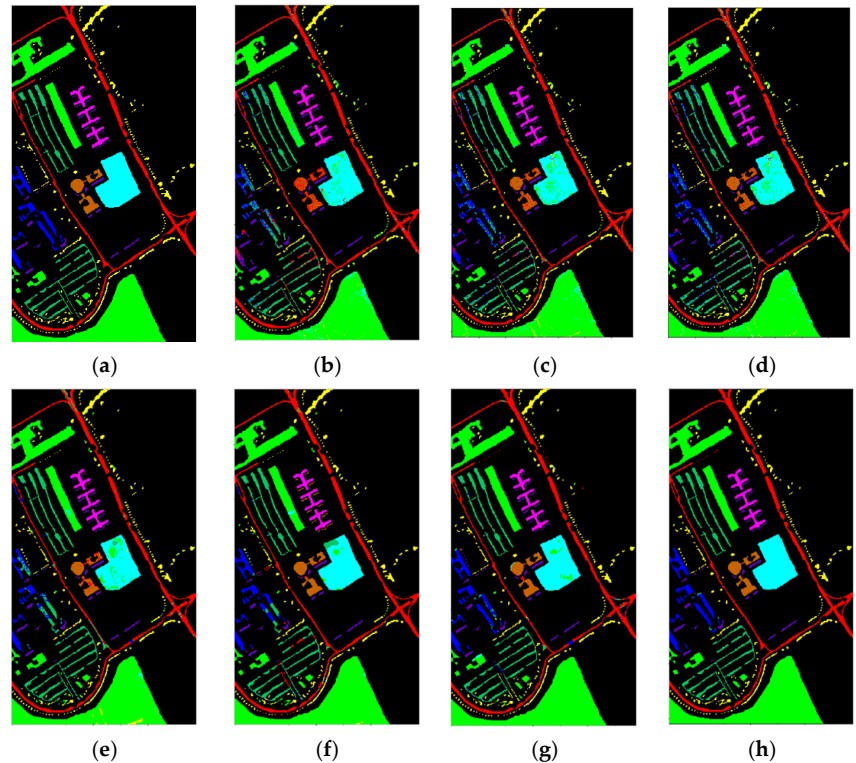

**Figure 5.** Visualization of the classification results on the PaviaU dataset. (**a**) Ground Truth. (**b**) RBF-SVM. (**c**) DBN. (**d**) SAE. (**e**) CNN. (**f**) 3DCNN. (**g**) PPF-CNN. (**h**) ARL-GAN.

**Table 6.** Classification comparison result of ARL-GAN with different algorithms for the PaviaU dataset.

| Method | RBF-SVM | DBN | SAE | CNN | 3DCNN | PPF-CNN | ARL-GAN |
|---|---|---|---|---|---|---|---|
| OA (%) | 89.3 ± 0.7 | 90.8 ± 0.6 | 91.5 ± 0.1 | 92.3 ± 1.2 | 97.9 ± 0.4 | 92.8 ± 0.4 | **99.97 ± 0.02** |
| AA (%) | 92.5 ± 0.6 | 94.5 ± 0.8 | 94.9 ± 0.2 | 94.2 ± 0.9 | 98.0 ± 0.6 | 95.5 ± 0.7 | **99.97 ± 0.01** |
| Kappa | 88.0 ± 0.8 | 89.7 ± 0.7 | 90.6 ± 0.4 | 91.4 ± 1.4 | 97.9 ± 0.5 | 91.9 ± 0.4 | **99.96 ± 0.02** |
| 1 | 97.4 ± 1.5 | 98.5 ± 0.8 | 97.9 ± 0.5 | 93.3 ± 8.7 | 99.4 ± 0.6 | 98.5 ± 0.5 | **100.0 ± 0.0** |
| 2 | 99.7 ± 0.2 | 98.9 ± 0.2 | 99.1 ± 0.5 | 97.4 ± 1.2 | 99.4 ± 0.6 | 99.7 ± 0.2 | **100.0 ± 0.0** |
| 3 | 93.7 ± 1.5 | 97.5 ± 0.1 | 95.3 ± 0.6 | 86.4 ± 4.1 | 98.8 ± 1.8 | 99.8 ± 0.1 | **100.0 ± 0.0** |
| 4 | 97.8 ± 1.3 | 99.0 ± 0.3 | 99.5 ± 0.6 | 98.2 ± 1.8 | 98.8 ± 1.9 | 99.7 ± 0.2 | **100.0 ± 0.0** |
| 5 | 97.5 ± 1.1 | 97.5 ± 0.2 | 98.5 ± 0.4 | 98.1 ± 1.0 | 99.0 ± 0.7 | 96.8 ± 0.2 | **100.0 ± 0.1** |
| 6 | 99.5 ± 0.3 | 99.3 ± 0.1 | 99.9 ± 0.1 | 99.9 ± 0.2 | 99.8 ± 0.3 | 99.8 ± 0.3 | **100.0 ± 0.0** |
| 7 | 99.3 ± 0.2 | 99.0 ± 0.3 | 99.2 ± 0.1 | 99.0 ± 0.9 | 97.9 ± 2.5 | 99.5 ± 0.2 | **99.9 ± 0.1** |
| 8 | 88.9 ± 2.9 | 83.0 ± 1.4 | 82.7 ± 0.7 | 88.4 ± 2.8 | 96.7 ± 2.2 | 89.9 ± 0.9 | **100.0 ± 0.0** |
| 9 | 99.2 ± 0.3 | 99.0 ± 0.1 | 99.2 ± 0.1 | 95.1 ± 0.7 | 99.8 ± 0.3 | 99.8 ± 0.2 | **100.0 ± 0.0** |
| 10 | 88.8 ± 1.9 | 92.8 ± 0.1 | 88.9 ± 0.9 | 93.6 ± 2.4 | 98.2 ± 2.1 | 88.3 ± 2.7 | **99.9 ± 0.1** |
| 11 | 87.5 ± 4.6 | 91.7 ± 0.1 | 93.6 ± 7.0 | 97.6 ± 1.0 | 97.3 ± 3.8 | 93.4 ± 2.9 | **100.0 ± 0.0** |
| 12 | 98.0 ± 2.3 | 99.0 ± 0.1 | 96.8 ± 1.0 | 98.9 ± 1.1 | 98.5 ± 2.1 | 99.7 ± 0.7 | **99.9 ± 0.1** |
| 13 | 98.0 ± 0.8 | 99.3 ± 0.2 | 99.2 ± 0.7 | 94.7 ± 2.4 | 97.4 ± 5.4 | 98.6 ± 0.7 | **100.0 ± 0.0** |
| 14 | 89.6 ± 2.6 | 92.0 ± 7.4 | 94.8 ± 0.2 | 92.5 ± 3.9 | 98.9 ± 1.0 | 92.3 ± 1.7 | **99.9 ± 0.2** |
| 15 | 53.9 ± 7.6 | 69.5 ± 0.2 | 75.9 ± 2.4 | 80.1 ± 5.3 | 96.5 ± 1.5 | 72.9 ± 2.5 | **99.8 ± 0.1** |
| 16 | 90.8 ± 5.0 | 96.1 ± 2.5 | 96.1 ± 1.9 | 93.6 ± 2.6 | 94.7 ± 5.1 | 95.7 ± 2.6 | **100.0 ± 0.0** |

The bold value indicates the optimal value.

Figure 6 is the visual classification result of different algorithms for the Salinas dataset. First, it can be seen that RBF-SVM, DBN, SAE, CNN, and PPF-CNN classify many of the eighth-class samples into fifteen categories, and many of the fifteenth-class samples into eighth categories. This is due to the problems of "synonyms spectrum" and "foreign body with the spectrum" in hyperspectral images. Compared with the five algorithms, 3DCNN has fewer points of error. This is because it uses the joint information of the spatial spectrum to reduce the impact of this problem. Compared with these algorithms, the algorithm in this paper can well distinguish between Class 8 and Class 15 samples, extract more distinguishing features, and achieve better classification results.

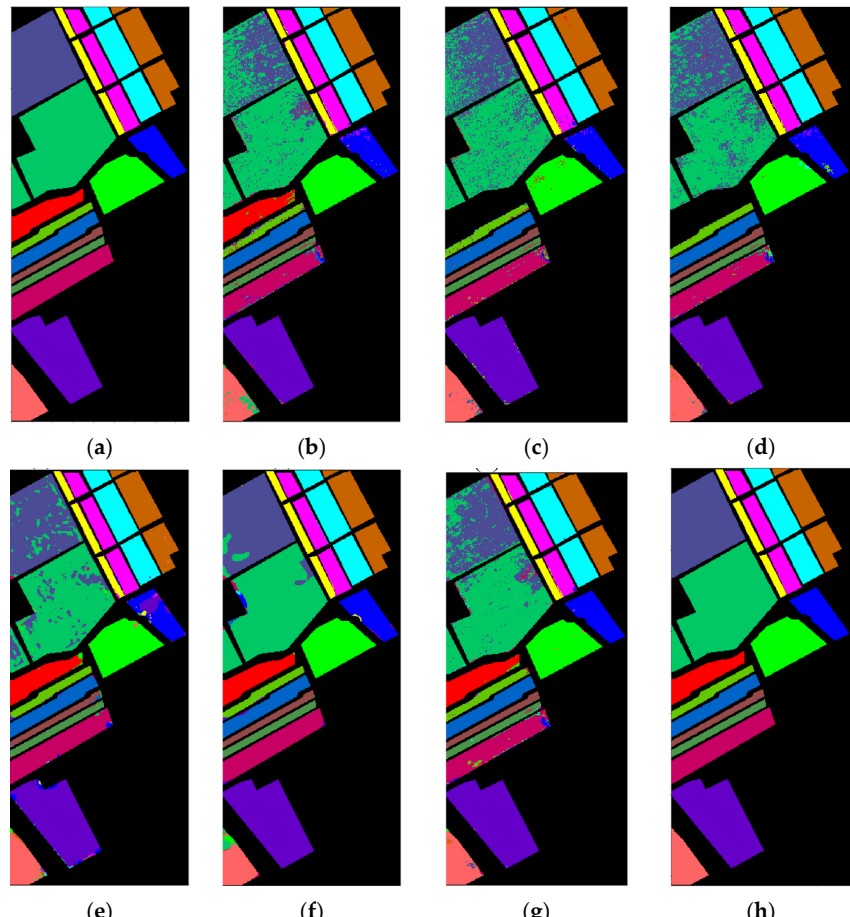

**Figure 6.** Visualization of the classification results for the Salinas dataset. (**a**) Ground Truth. (**b**) RBF-SVM. (**c**) DBN. (**d**) SAE. (**e**) CNN. (**f**) 3DCNN. (**g**) PPF-CNN. (**h**) ARL-GAN.

The above experimental results show that compared with 3DCNN, the algorithm in this paper generally has fewer misclassification points at the boundary, and, compared with PPF-CNN, the algorithm in this paper has fewer misclassification points inside the region. The reason for this is that 3DCNN uses 3D convolutional networks, and the network model needs to be trained with more labeled samples in order to obtain excellent classification performance while PPF-CNN extends the training dataset by reorganizing the existing training samples in pairs and relabeling them but ignores the use of unlabeled samples. However, due to the addition of Conditional Entropy and GAN, the algorithm in this paper makes full use of labeled samples and a large amount of unlabeled sample data to enhance the feature extraction ability of the encoder.

### 3.4. Effectiveness Analysis of Adversarial Learning and Conditional Entropy

In order to verify the effectiveness of the conditional entropy loss function and adversarial learning that is proposed by the algorithm in this paper, the classification results of

CNN, CNN-CE, and the algorithm in this paper are compared for the three datasets. The CNN classifiers in CNN and CNN-CE adopt the same network structure and consistent parameters as the classifiers in this paper. CNN-CE represents the algorithm model of semi supervised training with conditional entropy. Table 7 lists the classification results of the three algorithms on OA, AA, and Kappa coefficients. As shown in the table, compared with CNN and CNN-CE, the proposed method improved the OA by 0.42%, 0.89%, and 0.84%, respectively. It can be seen that semi-supervised training through conditional entropy can effectively improve the classification effect.

**Table 7.** Classification result of CNN, CNN-CE, and ARL-GAN.

| Dataset | Methods | CNN | CNN-CE | ARL-GAN |
|---|---|---|---|---|
| Indian Pines (Train: 5%) | OA | 97.04 ± 0.42 | 97.46 ± 0.43 | **98.25 ± 0.36** |
| | AA | 96.17 ± 0.93 | 93.63 ± 0.46 | **95.03 ± 2.07** |
| | Kappa | 96.62 ± 0.47 | 97.10 ± 0.49 | **98.01 ± 0.42** |
| Pavia University (Train: 3%) | OA | 98.89 ± 0.08 | 99.78 ± 0.06 | **99.83 ± 0.07** |
| | AA | 98.21 ± 0.23 | 99.55 ± 0.12 | **99.66 ± 0.18** |
| | Kappa | 98.53 ± 0.14 | 99.71 ± 0.11 | **99.77 ± 0.09** |
| Salinas (Train: 1%) | OA | 99.13 ± 0.13 | 99.96 ± 0.02 | **99.97 ± 0.02** |
| | AA | 98.27 ± 0.23 | **99.98 ± 0.01** | 99.97 ± 0.01 |
| | Kappa | 99.03 ± 0.09 | **99.97 ± 0.02** | 99.96 ± 0.02 |

The bold value indicates the optimal value.

As shown in the table, compared with CNN-CE, the algorithms in this paper have improved by 0.79%, 0.05%, and 0.01% respectively on the three datasets under the OA evaluation standard. The improvement rate of confrontation learning on the Indian dataset is relatively large, and the improvement is not large on the other two datasets because the training samples of the Indian Pines dataset are more limited, and the dataset is more noisy and more difficult to be classified so there is more room for improvement. The other two datasets have sufficient samples and are much easier to be classified, so there is less room for improvement. Through the overall analysis of the three datasets, the conditional entropy loss function and confrontation learning can improve the classification effect to a certain extent. The reason is that conditional entropy can constrain the classifier, train the network with a large number of unlabeled samples, and enhance the feature extraction ability of the network for hyperspectral data. Adversarial learning can improve both the generative performance of the generator and the discriminant ability of the discriminator, as well as the final classification performance.

*3.5. Number Sensitivity Analysis of Labeled Samples*

The method that is based on deep learning is highly dependent on labeled training samples, and the number of labeled training samples will greatly affect the classification performance. Therefore, it is necessary to study the sensitivity of the classification algorithm to the number of labeled training samples. Figures 7–9 show the overall classification accuracy (OA) of seven algorithms on three datasets: Indian Pines, PaviaU, and Salinas with different numbers of training samples; the horizontal coordinate is the number of labeled samples. In the experiment, the number of labeled training samples on the Indian Pines dataset ranged from 1% to 9% with an interval of 2%. The number of labeled training samples in the PaviaU dataset ranged from 1% to 5% with an interval of 1%. The number of labeled training samples in the Salinas dataset ranged from 1% to 3%, with an interval of 0.5%. For the Indian Pines dataset, the classification accuracy of the seven algorithms decreases significantly with the decrease of the number of training samples. However, compared with the other six algorithms, the algorithm in this paper has the lowest reduction rate, with 85% classification accuracy even when only 1% of the training samples are available. The other six algorithms have less than 60% classification accuracy except 3DCNN. The algorithm that is presented in this paper can effectively handle small

sample problems on the Indian Pines dataset. Among the six comparison algorithms, PPF-CNN, and 3DCNN achieve better classification results because they use the combination of sample pair and spatial-spectral to effectively reduce the impact of data volume on the classification results. The conclusion for the Pavia University dataset is basically the same as that for the Indian Pines dataset. The classification accuracy of the algorithms in this paper was more than 99% on the Salinas dataset, with no significant decrease. At the same time, the algorithm in this paper obtains optimal classification results on three datasets.

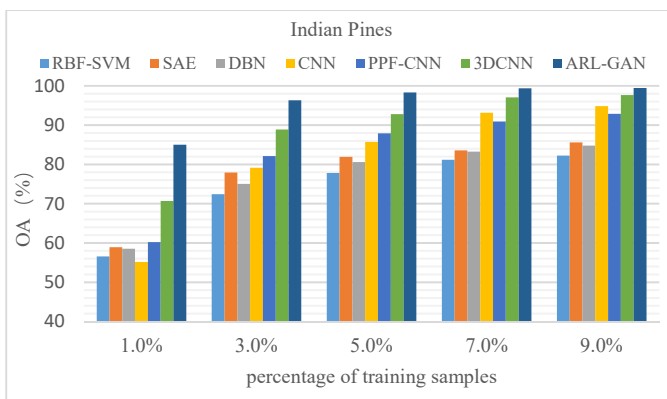

**Figure 7.** Accuracy of seven classification algorithms for the Indian Pines dataset under different numbers of training samples.

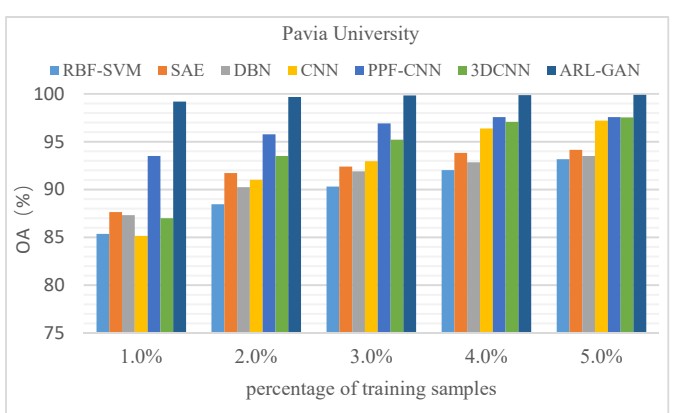

**Figure 8.** Accuracy of seven classification algorithms for the PaviaU dataset under different numbers of training samples.

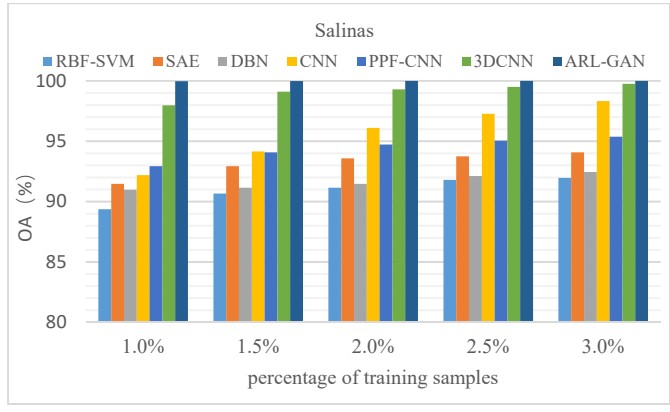

**Figure 9.** Accuracy of seven classification algorithms for the Salinas dataset under different numbers of training samples.

Therefore, the algorithm in this paper is a better choice when there are a limited number of labeled samples. This result is attributed to the full utilization of the large amount of unlabeled data by the algorithm in this paper and the excellent design of the classifier, which can obtain a good classification accuracy even when there are very few labeled samples.

### 3.6. Comparison with Existing Hyperspectral Image Classification Algorithms Based GAN

In this section, three generative adversarial networks that are based HSI classification methods are also introduced for comparison. The comparison results are shown in Table 8.

**Table 8.** Classification results of 3DGAN, HSGAN, MSGAN, and ARL-GAN.

| Dataset | Methods | HSGAN | 3DGAN | MSGAN | ARL-GAN |
|---|---|---|---|---|---|
| Indian Pines (5%) | OA | 74.14 | 95.13 | 95.61 | **98.25** |
| | AA | 65.63 | 84.81 | 91.44 | **95.03** |
| | Kappa | 70.44 | 94.89 | 95.02 | **98.01** |
| PaviaU (3%) | OA | 85.71 | 98.22 | 99.17 | **99.83** |
| | AA | 81.42 | 87.17 | 98.42 | **99.66** |
| | Kappa | 84.28 | 94.93 | 98.89 | **99.77** |
| Salinas (1%) | OA | 88.29 | 98.88 | 99.11 | **99.97** |
| | AA | 92.31 | 92.61 | 99.23 | **99.97** |
| | Kappa | 87.03 | 97.92 | 99.03 | **99.96** |

The bold value indicates the optimal value.

3DGAN [18] uses image blocks as training samples and uses antagonistic learning to improve the discriminator's classification ability. This paper also uses image blocks as training samples, but uses the antagonistic network to perform representation learning and uses the features that are extracted by the encoder to classify. The algorithm in this paper improves the OA index of Indian Pines, PaviaU, and Salinas datasets by 31.5%, 1.63%, and 1.07%, respectively, compared with 3DGAN. HSGAN [22] uses one-dimensional spectral samples as training samples, uses GAN to extract spectral features with discriminators in an unsupervised manner, trains classifiers in a supervised manner, and finally votes for the categories of the test samples from their neighboring cells. Compared with the HSGAN algorithm in this paper, OA improves by 24.15%, 14.13%, and 11.67% on three datasets, respectively. Using image blocks as training samples can greatly improve the classification accuracy of the algorithm. MSGAN uses two generators and a discriminator to form a variety of spatial spectrum generation adversarial networks hyperspectral image classification. The two generators generate spectral samples and spatial image block samples respectively. The discriminator fuses the spatial spectrum features to improve the discriminator's classification performance in an end-to-end manner. MSGAN uses spatial spectrum information and is more accurate than 3DGAN and HSGAN. Although the algorithm in this paper also utilizes spatial and partial spectral information, the classification effect is better than MSGAN, and the OA is improved by 2.65%, 0.63%, and 0.87% for the three datasets, respectively, which shows that the algorithm in this paper can improve the classification accuracy more effectively by using the GAN approach. The reason is that the algorithm adds the encoder as the final classifier and modifies the discriminator to guide the feature extraction of the encoder, which is more suitable for the classification task than the previous algorithms that are based on GAN that uses the discriminator as the final classifier. In addition, with the addition of conditional entropy, the algorithm makes full use of the information of unlabeled samples, so it is more suitable for improving the problem of small samples.

### 4. Conclusions

In this paper, for the problem of small samples in hyperspectral image classification tasks, an HSI classification algorithm that is based on adversarial representation learning is

proposed, which introduces unlabeled samples and uses conditional entropy and GAN for semi-supervised learning. The algorithm also extending GAN to the field of representation learning, and uses adversarial learning to improve the ability of the encoder to extract features that are more suitable for the classification task. In the experimental part of this paper, the effectiveness of the algorithm is verified by experiments. For the three datasets, the classification accuracy that is obtained by the algorithm is 98.3%, 99.8%, and 99.9%, respectively. Compared with the optimal results of other algorithms, the algorithm in this paper increases by 5.5%, 2.9%, and 2.0%, respectively. Conditional entropy is used as constraint to enhance the performance of the classifier. However, conditional entropy also increases the error prediction of classifier. The proposed algorithm reduces the adverse effects of conditional entropy by manually-designed step weights. In our future work, we will try to select appropriate weights adaptively and adopt other regularization constraints. In the process of training, the model training speed of this method is relatively slow.

**Author Contributions:** S.Z. wrote the whole article and adjusted it. X.Z. directed the completion of the algorithm and put forward key suggestions. T.L. adjusted the structure of the article and checked out some mistakes. H.M. reviewed the article and made some suggestions. X.C. reviewed the whole paper carefully and gave some key suggestions on the format and writing style of the paper. L.W. checked out some grammatical errors in the paper. All authors have read and agreed to the published version of the manuscript.

**Funding:** This work was supported in part by the National Natural Science Foundation of China under Grant 61877066, Aero-Science Fund under Grant 20175181013 and Science and technology plan project of Xi'an under Grant 2021JH-05-0052.

**Data Availability Statement:** The data presented in this study are available in this article.

**Conflicts of Interest:** The authors declare no conflict of interest.

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
