# Peer review of "Adversarial Representation Learning for Hyperspectral Image Classification with Small-Sized Labeled Set"

_remotesensing, doi:10.3390/rs14112612_

Round 1

Reviewer 1 Report

  1. Abstract: The “Abstract” section can be made much more impressive by highlighting your contributions. The contribution of the study should be explained simply and clearly.
  2. Introduction: The proposed solution to the problem in the introduction section is not clearly explained. It needs proper attention of the authors and states the limitations of existing systems and clear solution of your system to highlight the main influence of this work on over-employed techniques.
  3. The current challenges are not crystal clearly mentioned in the introduction section too.
  4. Please discuss your findings and what the advantage of your proposed model is.
  5. Table 1 is vague and needs more explanation to show the step-wise data flow clearly.
  6. Equations 9 and 10 description is different from than actual computation the authors need to clarify and should explain accordingly.
  7. Experiment: Authors are suggested to include more explanation and discussion on the results analysis of their proposed methods. Authors should discuss and analyze the results based on their real actual numerical values in tables and the performance of their proposed methods. The analysis of results should justify the advantages of their proposed methods in comparison to other methods. Meanwhile, the authors also have to provide some insightful discussion of the results.
  8. Section Conclusion - Authors are suggested to include in the conclusion section the real actual results for the best performance of their proposed methods in comparison towards other methods to highlight and justify the advantages of their proposed methods.
  9. The manuscript, however, does not link well with recent literature on deep learning that appeared in relevant top-tier journals, such as 1D-CNN: Speech Emotion Recognition System Using a Stacked Network with Dilated CNN Features”.
  10. Future direction is not mentioned it should be added in detail in the revised version at the end of the conclusion section.
  11. The linguistic quality needs improvement. It is essential to make sure that the manuscript reads smoothly- this definitely helps the reader fully appreciate your research findings.
  12. What is the main difficulty when applying the proposed method? The authors should clearly state the limitations of the proposed method in practical applications and should be mentioned in the article's conclusion.

Author Response

Dear anonymous Reviewer:

We would like to express our sincere gratitude to you for your time and constructive suggestions. Your valuable comments help us improve the quality and clarity of the manuscript. All the comments have been seriously considered and addressed in the revised manuscript. In the following, we will explain the answers to the reviewers’ comments one by one. The red sentences are our replies, and we highlight the modified parts in yellow.

We would like to express our sincere appreciation to you for your valuable comments one more time, and hope that the new version will meet with approval.

Yours sincerely,

Shuhan Zhang, Xiaohua Zhang, Tianrui Li, Hongyun Meng, Xianghai Cao, Li Wang

Reviewer 2 Report

The manuscript introduces a new variant of DN methods to explicitly face the problem of small size training data set in Hyperspectral remotely sensed images, an always active field. Even though general ideas in the manuscript cannot be considered pioneeristic, however authors propose interesting variations on the theme that apparently improve performance of some of the best known competitor methods on well known data sets. Therefore the manuscript can be considered for possible publication in the Journal.   I have a general doubt, solicited by the practically perfect performance on one or two data sets. The mandatory paradigma for evaluating performances is that hyperparameters have to be trained on a different data set over which performance indicators are evaluated. In a rigorous framework 3 data sets are considered (Training+Validation for estimate of hyperparameters and Test for the final evaluation of Indicators). In this manuscript (but common also to other ones in the literature) the Training data set is apparently chosen pointwise and randomly inside an image. For the very nature of the network structure of the DN methodology, blocks around such random pixels are considered. Due to the (relatively) small size of the considered images, the mosaic of such Training blocks covers a large part of the image (if not the entire one), de facto violating the principle of separate Training and Validation/Test data sets. In an ideal case, also to remove local correlation effects from the evaluation of the performance, 2 different and disjoint images (or subimages of a same image) should be considered as a Training/Validation as a Test, respectively.   I ask authors to clarify the following items:   Section 2.2, l. 206-212: Apparently a nonlinear activation function is included in the Discrimination function. Considering that some activation functions can by steep (when not discontinuous), how authors justify accuracy of the linear approximation? Section 3.1: How is training data set obtained? Since the method works by blocks of images, what do author mean by 1%, 3% and 5% sampling? Of course it is intended by pixel, because there are no blocks enough due to the small/medium size of images. Therefore how do they deal with blocks from pixels? Are bocks surrounding the chosen pixels? In this case there are probably overlapping blocks. This problem is important in my opinion, because the set of blocks in the training data set can cover a large part of the image (if not the entire data set), and therefore the hypothesis of separate training and test data sets is violated, which could justify some almost perfect claims (as the practically perfect classification on the Pavia data set).  In addition, are pixels selected purely randomly or stratified by class? Some classes are significantly less populated. Finally, what do authors mean by “from” in “samples were selected from training”? Do they intend from the image?   Sections 3-4 and 3.6. it is not clear the objective. It looks like that comparison of Performance indicators is provided for the proposed method in comparison with two methodologies (CNN and CNN-CE) 3 GAN ones, respectively.. In other words, exactly the same comparison shown in Section 3.3 (and Tables 5-7). If this is the case, I would suggest to merge the three Sections (and corresponding Tables). Please also note that the same notation (CNN) for two different methods generates confusion.   AA, OA indicators are never explained nor introduced (the same for Kappa)   Table 6: please put boldface only the best results among the methods   Minor remarks:   l. 66-68: something is wrong in the sentence (the dot after [24] l. 83-84: “similar input in- 83 stances should be similar output labels are shared,” sentence unclear l. 91-92: “The samples generation by GAN do not need to 91 give the sample distribution in advance” sentence unclear l. 123-124: “This makes the obtained feature extractor be better 123 suited for classified tasks” sentence is unclear l. 127-128: “whose input noise is replaced by the feature vector extracted by the encoder” sentence unclear, please explain better, one does not replaces noise by features l. 141: I think authors intend ARL l. 195-196: “in terms of "layer parameters" in terms of spectral parametrization” please rephrase l. 202: please replace “equation (1). Where” by “equation (1), where” l. 215: remove “there is” l. 221: “decomposition” instead of “decom-position” l. 273: “In (13) the first term” instead of “Formula (13) The first term” l. 302: Please leave a space after dot l. 311-317: Sentence too long, unclear l. 329: remove comma from “is,” l. 359: Please check the size of the image, it does not seem to correspond to original data (and also to the images shown in the manuscript) l. 394: remove dot after blocks l. 521: “Through the above comparison experiments on three data and we can see” please rephrase l. 601: “Classification” instead of “Classi-fication”

Author Response

(The authors gave the same response as above.)

Round 2

Reviewer 1 Report

The authors successfully addressed my comments and suggestions. Good Luck!